# Symmetry Control Neural Networks

## Abstract

This paper continues the quest for designing the optimal physics bias for neural networks predicting the dynamics of systems when the underlying dynamics shall be inferred from the data directly. The description of physical systems is greatly simplified when the underlying symmetries of the system are taken into account. In classical systems described via Hamiltonian dynamics this is achieved by using appropriate coordinates, so-called cyclic coordinates, which reveal conserved quantities directly. Without changing the Hamiltonian, these coordinates can be obtained via canonical transformations. We show that such coordinates can be searched for automatically with appropriate loss functions which naturally arise from Hamiltonian dynamics. As a proof of principle, we test our method on standard classical physics systems using synthetic and experimental data where our network identifies the conserved quantities in an unsupervised way and find improved performance on predicting the dynamics of the system compared to networks biasing just to the Hamiltonian. Effectively, these new coordinates guarantee that motion takes place on symmetry orbits in phase space, i.e. appropriate lower dimensional sub-spaces of phase space. By fitting analytic formulae we recover that our networks are utilising conserved quantities such as (angular) momentum.

## 1 Introduction

Building in a bias to neural networks has been a key mechanism to achieve extra-ordinary performance in tasks such as classification. A standard example is to utilise translation invariance in convolutional neural networks Krizhevsky et al. (2012) and by now building in equivariance to other symmetries such as rotational symmetries has proven to be very successful (e.g. Cohen & Welling (2016)).

Possible motions are constrained due to symmetries of the system. In technical terms, motion takes place on a subspace of phase space. Energy conversation – related to invariance under time translation – has been utilised in the context of Hamiltonian Neural Networks (HNNs) Greydanus et al. (2019) where the energy functional, i.e. the Hamiltonian is inferred from data. This approach has seen large improvements in predicting the dynamics over baseline neural networks which simply try to predict the change of phase-space coordinates in time. Here we extend this approach by learning and incorporating additional constraints due to further symmetries of the system.

Coarsely speaking, finding symmetries corresponds to finding good coordinates. In classical mechanics this is achieved by performing canonical transformations and identifying cyclic coordinates which reveal conserved quantities. The aim of this paper is to demonstrate that multiple conserved quantities can indeed be automatically found in this way which has not been demonstrated beforehand.

Similar in spirit to learning the Hamiltonian, we formulate loss functions which enforce a representation in terms of cyclic coordinates and use them as the input for our Hamiltonian, differing from previous flow-based approaches searching for these coordinates (Bondesan & Lamacraft, 2019; Li et al., 2020). We experimentally find as a proof of principle that this mechanism identifies the underlying conserved quantities such as angular momentum, momentum, the splitting into decoupled subsystems, and can find the number of conserved quantities.

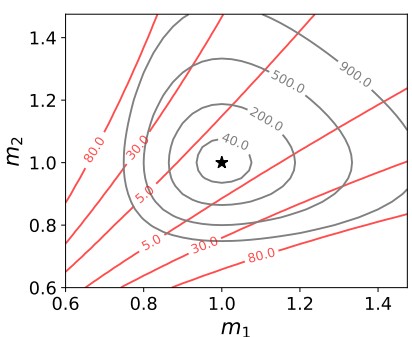

Figure 1: **Effect of additional loss components:** Loss-contours for the HNN-loss (6) (shown in gray) and the Poisson loss (7) (red) arising from the angular momentum ($\|\{L, H\}\|_2$) in the 2-body Hamiltonian (10) with respect to two model parameters $m_1$ and $m_2$. The data model corresponds to $m_1 = m_2 = g = 1$ (indicated with a star) and we evaluate the loss over our training set. The analytic constraint $m_1 = m_2$ which arises from evaluating the Poisson bracket $\{L, H\} \sim (m_1 - m_2)$ is clearly visible and provides additional constraints on the model parameter space.

We demonstrate significant improvement in the predictions of the underlying Hamiltonian and subsequently the dynamics of the system. From our trained networks, we can find analytic expressions for the conserved quantities and determine the number of conserved quantities.

## 2 THEORY

We briefly describe the standard techniques in Hamiltonian mechanics which our network utilises.[1] We consider a classical system with $N$ particles in $d$ spatial dimensions. Such a system can be described by the variables $(\mathbf{q}, \mathbf{p})$, where $\mathbf{q} = (q_1, ..., q_{N \cdot d})$ are typically the positions for each dimension of the objects and $\mathbf{p} = (p_1, ..., p_{N \cdot d})$ are the corresponding momenta. This is the input to our network and we are interested in predicting the time-evolution of this system, i.e. $(\mathbf{q}, \mathbf{p})$ at later time steps.

This pair $(\mathbf{q}, \mathbf{p})$ is an element of phase space in which every point corresponds to a state the system can take. The time evolution is governed by the Hamiltonian $\mathcal{H}(\mathbf{q}, \mathbf{p})$[2] and the associated Hamiltonian equations:

$$\frac{d\mathbf{q}}{dt} = \frac{\partial \mathcal{H}}{\partial \mathbf{p}} = \{\mathbf{q}, \mathcal{H}\}, \qquad \frac{d\mathbf{p}}{dt} = -\frac{\partial \mathcal{H}}{\partial \mathbf{q}} = \{\mathbf{p}, \mathcal{H}\}, \tag{1}$$

where $\{\bullet, \bullet\}$ are the Poisson bracket. They are defined as

$$\{f, g\} := \sum_{i=1}^{N \cdot d} \frac{\partial f}{\partial q_i} \frac{\partial g}{\partial p_i} - \frac{\partial f}{\partial p_i} \frac{\partial g}{\partial q_i}. \tag{2}$$

The Poisson bracket does not only arise for the time evolution of the canonical coordinates $(\mathbf{q}, \mathbf{p})$ but also for any function of these coordinates $g(\mathbf{q}, \mathbf{p})$ which does not explicitly depend on time:

$$\frac{dg(\mathbf{q}, \mathbf{p})}{dt} = \sum_{i=1}^{N \cdot d} \frac{\partial g}{\partial q_i} \frac{dq_i}{dt} - \frac{\partial g}{\partial p_i} \frac{dp_i}{dt} = \{g, \mathcal{H}\}, \tag{3}$$

where we have used the Hamiltonian equations (1) in the last step. From this expression we see that the Poisson bracket of a conserved quantity with the Hamiltonian $\mathcal{H}(\mathbf{q}, \mathbf{p})$ vanishes and that the Hamiltonian itself is a conserved quantity.

The physics of this system is invariant under diffeomorphic coordinate transformations on the canonical coordinates and we will use coordinate transformations to reveal the constants of motions and hence the symmetries of the system. We use a particular type of diffeomorphic transformations $T$, namely *canonical transformations* which are transformations that leave the structure of the Hamiltonian equations (1) and in particular the Poisson bracket unchanged:

$$T : (\mathbf{q}, \mathbf{p}) \mapsto (\mathbf{Q}(\mathbf{q}, \mathbf{p}), \mathbf{P}(\mathbf{q}, \mathbf{p})),$$
$$\{f, g\}_{\mathbf{p}, \mathbf{q}} = \{f, g\}_{\mathbf{P}, \mathbf{Q}}, \ \mathcal{H}(\mathbf{p}, \mathbf{q}) = \tilde{\mathcal{H}}(\mathbf{P}(\mathbf{p}, \mathbf{q}), \mathbf{Q}(\mathbf{p}, \mathbf{q})). \tag{4}$$

---

[1]We refer the reader for more details to standard textbooks such as Landau & Lifshitz (1982).
[2]We focus on time-independent Hamiltonians for simplicity.

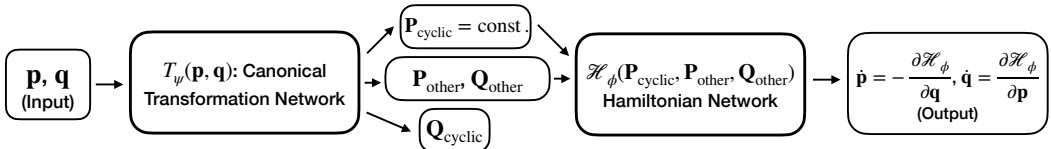

Figure 2: Structure of symmetry control networks: The network $T_\psi$ transforms the input coordinates of phase space $(\mathbf{p}, \mathbf{q})$ to canonical coordinates where some coordinates are forced to be cyclic. These coordinates are used as input to the Hamiltonian network $H_\phi$. The output, the time derivatives of our initial coordinates, is calculated from the Hamiltonian using auto-differentiation.

We are interested in finding transformations $T$ such that at least one coordinate satisfies

$$0 = \dot{P}_i = -\frac{\partial \mathcal{H}}{\partial Q_i} = \{P_i, \mathcal{H}\}. \tag{5}$$

Such a coordinate $P_i$ is conserved in the system and the Hamiltonian does not depend on the associated $Q_i$, i.e. it depends on fewer degrees of freedom and the motion in phase space is restricted to a lower dimensional manifold. Put differently, the cyclic coordinates provide via constraints of the type (5) additional restrictions on the allowed Hamiltonian function which we learn with our symmetry control neural networks (cf. Figure 1 shows explicit constraint from angular momentum conservation in a 2-body example in addition to constraints arising from satisying the Hamiltonian equations of motion).

**Symmetry Control Neural Networks:** In a first step we search for such *cyclic coordinates* with a network $T_\psi$ and use them as input for our Hamiltonian $\mathcal{H}_\phi$. The structure of our two trainable networks is shown in Figure 2. In order to find cyclic coordinates and to identify a Hamiltonian, our loss function contains several components:

1. The first loss ensures that our Hamiltonian satisfies Hamiltonian equations (1), which we can ensure as follows:

$$\mathcal{L}_{\text{HNN}} = \sum_{i=1}^{N \cdot d} \left\| \frac{\partial \mathcal{H}_\phi(\mathbf{P}, \mathbf{Q})}{\partial p_i} - \frac{dq_i}{dt} \right\|_2 + \left\| \frac{\partial \mathcal{H}_\phi(\mathbf{P}, \mathbf{Q})}{\partial q_i} + \frac{dp_i}{dt} \right\|_2. \tag{6}$$

   The time derivatives are provided by the data and the derivatives of the Hamiltonian with respect to the input variables can be obtained using auto-differentiation. This is the same loss as introduced in Greydanus et al. (2019).

2. To ensure that our transformation $T_\psi$ are of the type we are interested in (cf. Eq. (4)), i.e. our new coordinates fullfil the Poisson algebra, we enforce the following loss:

$$\mathcal{L}_{\text{Poisson}} = \sum_{i,j=1}^{N \cdot d} \|\{Q_i, P_j\} - \delta_{ij}\|_2 + \sum_{i,j>i}^{N \cdot d} \|\{P_i, P_j\}\|_2 + \|\{Q_i, Q_j\}\|_2, \tag{7}$$

   where in some practical applications we only enforce this loss on $n$ cyclic coordinate pairs. The first part of this loss ensures that a vanishing solution is not allowed.

3. Hamilton's equations have still to be satisfied with respect to the new coordinates. For the cyclic coordinates we have enforced by our architecture that $\mathcal{H}_\phi$ is independent of $Q_i$. However, to ensure that $P_i$ is actually conserved, we require the following additional loss:

$$\mathcal{L}_{\text{HQP}}^{(n)} = \sum_{i=1}^{n} \left\| \frac{dP_i}{dt} \right\|_2 + \left\| \frac{dQ_i}{dt} - \frac{\partial \mathcal{H}_\phi(\mathbf{P}, \mathbf{Q})}{\partial P_i} \right\|_2$$
$$+ \beta \sum_{i=n+1}^{N \cdot d} \left\| \frac{dP_i}{dt} + \frac{\partial \mathcal{H}_\phi(\mathbf{P}, \mathbf{Q})}{\partial Q_i} \right\|_2 + \left\| \frac{dQ_i}{dt} - \frac{\partial \mathcal{H}_\phi(\mathbf{P}, \mathbf{Q})}{\partial P_i} \right\|_2, \tag{8}$$

   where $n$ denotes the number of cyclic variables we are imposing and $\beta$ denotes a hyperparameter for our experiments. The time derivatives can be calculated using either expressions in (3).

Our total loss is a weighted sum of these three components:

$$\mathcal{L} = \mathcal{L}_{\text{HNN}} + \alpha_1 \mathcal{L}_{\text{Poisson}} + \alpha_2 \mathcal{L}_{\text{HQP}}^{(n)} \, , \tag{9}$$

where the weights $\alpha_i$ are tuned.

For integrating the solutions in time from our respective symmetry control network we use a fourth order Runge-Kutta integrator as in Greydanus et al. (2019) which unlike symplectic integrators allows for a comparison with neural network approaches directly predicting the dynamics of a system.[3]

## 3   EXPERIMENTS

Our experiments are designed with the following goals in mind:

1. **SCNN-base:** We want to compare the performance of symmetry control neural networks with HNNs and baseline neural networks which directly predict $(\dot{\mathbf{q}}, \dot{\mathbf{p}})$. In these experiments we generally use the maximal number of conserved quantities. To evaluate the performance, we use the accuracy of the prediction and whether physically conserved quantities are actually conserved.

2. **SCNN-constraint:** We explore whether imposing domain knowledge about symmetries improves the performance. This is motivated by the fact that we often know about the existence of certain conserved quantities.

3. **Analytic formulae:** We evaluate whether our networks actually use conserved quantities in close proximity to known physically conserved quantities. To do this we fit analytical formulae on the conserved quantities and check how accurately we can recover known formulae.

Throughout, we run a hyperparameter analysis on how the different loss components influence our results to provide a first numerical analysis of these constraints. We test our method on synthetic examples and freely available experimental data, several of which have already been used in Greydanus et al. (2019).

### 3.1   PHYSICAL SYSTEMS AND OBJECTIVE OF EXPERIMENTS

We consider the **two-body problem** in two dimensions which is governed by the following Hamiltonian

$$\mathcal{H} = \frac{p_{x1}^2}{2m_1} + \frac{p_{y1}^2}{2m_1} + \frac{p_{x2}^2}{2m_2} + \frac{p_{y2}^2}{2m_2} - \frac{g}{\|\mathbf{q}_1 - \mathbf{q}_2\|_2} \, . \tag{10}$$

To simplify the problem we set $m_1 = m_2 = g = 1$. We generate datasets with near circular orbits where the whole system has a small velocity in an arbitrary direction to avoid accidental symmetries (see Appendix B for more details). Besides the Hamiltonian itself, conserved quantities are the total momentum in $x$ and $y$ direction, and the angular momentum:

$$P_x = p_{x1} + p_{x2} \, , \qquad P_y = p_{y1} + p_{y2} \, ,$$
$$L = (p_{x1} - p_{x2})(q_{y1} - q_{y2}) - (p_{y1} - p_{y2})(q_{x1} - q_{x2}) \, . \tag{11}$$

The second system is the **3-body system** which is governed by the following Hamiltonian:

$$\mathcal{H} = \sum_{i=1}^{3} \frac{p_{xi}^2}{2m_i} + \frac{p_{yi}^2}{2m_i} - \frac{1}{2} \sum_{i=1, j\neq i}^{3} \frac{g}{\|\mathbf{q}_i - \mathbf{q}_j\|_2} \, , \tag{12}$$

---

[3]We find that the main numerical inaccuracy in the prediction arises from inaccuracies of the Hamiltonian $\mathcal{H}_\phi$ rather than the choice of this integrator when comparing it with standard symplectic integrators (Rein & Liu, 2012). Here in this paper, our focus is on predicting the Hamiltonian most accurately rather than making most accurate predictions with a Hamiltonian.

where we take a univeral mass and coupling $m_i = g = 1$. As in the 2-body example, we consider systems which have a small velocity in an arbitrary direction (cf. Appendix B). Again, the overall momentum in both directions and the angular momentum are conserved quantities:

$$P_x = p_{x1} + p_{x2} + p_{x3} \ , \qquad P_y = p_{y1} + p_{y2} + p_{y3} \ ,$$
$$L = (2q_{x1} - q_{x2} - q_{x3}) \, p_{y1} - (2q_{y1} - q_{y2} - q_{y3}) \, p_{x1} + \text{even permutations} \ . \qquad (13)$$

The third system which we are interested in is the **n-dimensional coupled oscillator** which is described by the following Hamiltonian

$$\mathcal{H} = \sum_{i=1}^{n} \frac{p_i^2}{2m} + \sum_{i,j=1}^{n} q_i A_{ij} q_j \ , \qquad (14)$$

where we use $m = 1/2$ and the coupling matrix $A_{ij}$ is a symmetric $n \times n-$matrix with positive eigenvalues. As this matrix is not diagonal, the potential energy of each oscillator in these coordinates is influenced by the position of the other oscillators. By diagonalization the system can be described by decoupled harmonic oscillators. The conserved quantities are the energies of these subsystems:

$$E_i = \tilde{p}_i^2 + \lambda_i \tilde{q}_i^2 \ , \qquad (15)$$

where $\tilde{p}$ and $\tilde{q}$ are the canonical coordinates associated to the respective eigenvalue $\lambda_i$ of the coupling matrix $A_{ij}$. Note that these systems include spring-like interactions.

To include a model with few conserved quantities and to show that the SCNN can be used to determine the number of conserved quantities, we consider the **double pendulum**, a chaotic system, where only energy is conserved. We test the performance of SCNNs on a **charged massive particle in a background magnetic field** where the kinetic and potential energy are not decoupled in the Hamiltonian. To estimate the feasibility of our approach with experimental data where energy is not conserved, we test SCNNs on data from two experiments presented in (Schmidt & Lipson (2009)): two-masses-three-springs is a **coupled two-dimensional harmonic oscillator**, and we combine data from the physical pendulum to a **real spherical pendulum**. Details about these systems can be found in Appendix A and B.

## 3.2 RESULTS

### 3.2.1 2-BODY PROBLEM

We have trained our networks with data from 800 trajectories with 50 datapoints each and checked the performance on 200 different trajectories with 50 datapoints (cf. Appendix B for more details). We tested a large set of different hyperparameters (cf. Appendix C), where the large bulk of the models converges and reveals the conserved quantities. The implementation of our experiments is submitted alongside with this paper. We compare the following networks which all converge to a reasonable accuracy and our large training set ensures that overfitting does not pose a problem here:

- The **baseline architecture** has two hidden dense layers with 200 hidden units and tanh activation. The output is directly $(\dot{\mathbf{q}}, \dot{\mathbf{p}})$.

- **HNN:** We use the original architecture with two hidden dense layers, both networks using tanh activations and 200 hidden units.

- **SCNN-base:** we use a transformation network architecture with two hidden dense layers (tanh activation) and a Hamiltonian neural network with two hidden dense layers as above. We enforce four cyclic coordinates (the maximum possible number). As described below, we vary the respective loss-weights and find that a loss with $\alpha_{1,2} = 0.001$ works best.

- **SCNN-constraint:** we impose domain knowledge about conserved momenta $(P_x, P_y)$ and angular momentum $(L)$ respectively. The Hamiltonian is calculated with our

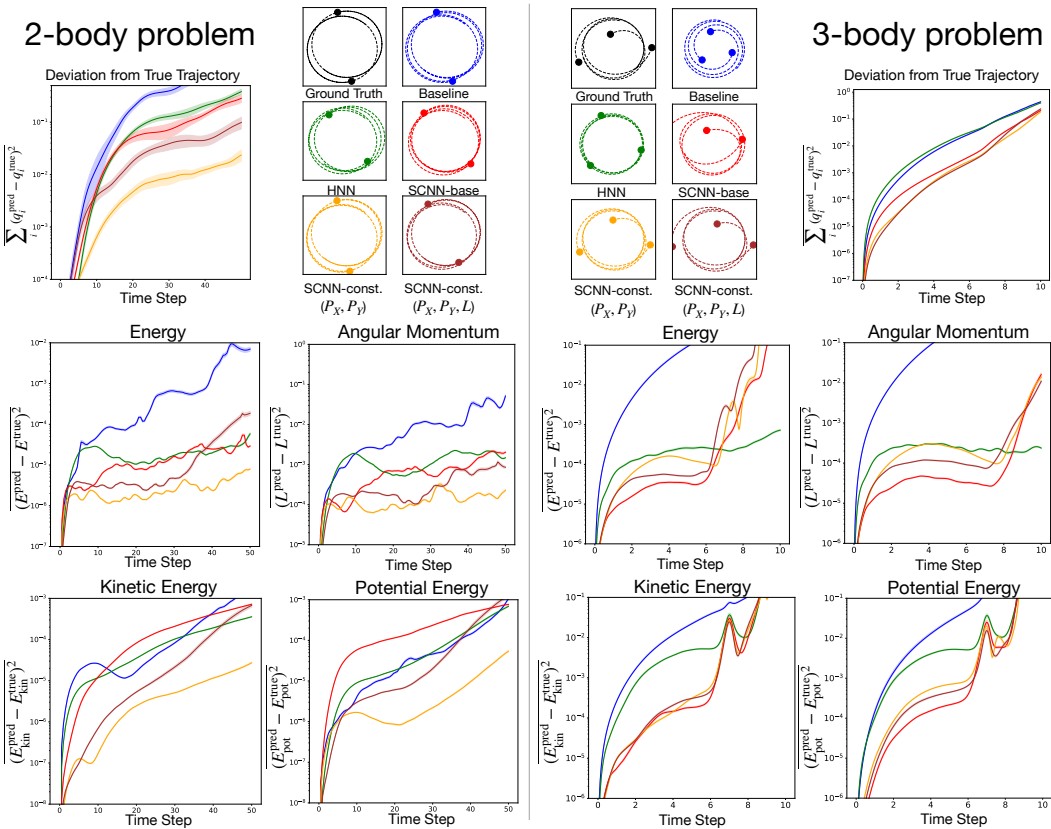

Figure 3: Performance overview of the two-body and three-body experiments. Top line includes sample trajectories for the ground truth, the baseline neural network, a HNN, and our symmetry control networks. The colour coding is such that the same colour is used throughout all plots in this figure. The top line also shows the deviation of the predicted from the true trajectory. The solid line denotes the mean deviation over 50 trajectories and the one sigma band is shown shaded around it. **Bottom two lines:** We show mean squared deviation from conserved quantities where again we show the mean deviation over 50 trajectories and and the associated one sigma band.

standard two hidden dense layers from the input $(\mathbf{q}, \mathbf{p})$. Beyond the HNN-loss, we only impose the loss components ensuring $\{H, P_i\} = 0$ and $\{H, L\} = 0$ respectively (cf. Figure 1 for a visualization). *Improve on loss description, if we want to keep it.*

For all networks we analyse the time evolution predicted from our trained network by determining the trajectory, the deviation from the ground truth and the evolution of the conserved quantities. A sample for the predicted trajectories is shown for each network in Figure 3. To quantify the accuracy of the respective networks we show the mean-squared deviation from the ground truth trajectories which we obtain over 50 different trajectories on our trained networks. We see a modest improvement with our SCNN-base architecture and an order of magnitude improvement when we include domain knowledge about conserved quantities for SCNN-constraint networks. Finally the bottom line of plots in Figure 3 shows the deviation of conserved quantities predicted from our networks in comparison to the underlying conserved quantity where we find an improvement compared to the HNN network.

To analyse the conserved quantities, we fit polynomials in $(\mathbf{q}, \mathbf{p})$ to the output of the cyclic coordinates. Picking the solution with the lowest degree and good accuracy[4] we recover known conserved quantities of this system in the **SCNN-base**-model:

$$
\begin{aligned}
P_{c1} = &- 4.21\ p_{x1} - 4.21\ p_{x2} - 1.26\ p_{y1} - 1.29\ p_{y2} \qquad (0.03)\ , \\
P_{c2} = &- 0.93\ p_{x1} - 0.92\ p_{x2} - 3.23\ p_{y1} - 3.22\ p_{y2} \qquad (0.03)\ , \\
L = &- 1.07\ q_{x1}p_{y1} + 0.88\ q_{x1}p_{y2} + 0.93\ q_{x2}p_{y1} - 1.03\ q_{x2}p_{y2} \\
&+ 1.01\ q_{y1}p_{x1} - 0.89\ q_{y1}p_{x2} - 0.92\ q_{y2}p_{x1} + 0.99\ q_{y2}p_{x2} \qquad (0.10)\ .
\end{aligned} \qquad (16)
$$

where the remaining terms are smaller than 0.1 and the number in brackets denotes the respective mean-squared error of the fit. The first two are clear linear combinations of the conserved momenta in Equation (11) and the third one is corresponding to angular momentum. To find these analytic fits we have simply used a polynomial ansatz and checked that there are no significant changes when allowing for higher powers in the analytic ansatz (cf. Appendix E).

### 3.2.2 THREE-BODY PROBLEM

Apart from adapting the input and output dimensions appropriately, we keep the baseline and HNN architecture unchanged to the two-body problem. For our symmetry control networks we report results for a **SCNN-base** architectures with six cyclic coordinates and **SCNN-constraint** architectures where we use our domain knowledge and enforce as in the two-body examples the conservation of momenta and angular momentum.

We train our networks starting again at near circular orbits (cf. Appendix B). However, after an initial circular motion for around 10 time steps the motion changes drastically when two particles come close to each other. In our experiments, we are not able to capture these dynamics and we do not aim to predict the dynamics after this initial drastic deviation. We observe in our experiments that our symmetry control networks are capable to predict this initial drastic change correctly. This is in contrast to the HNN and baseline networks whose predictions for the trajectories are worse than for the symmetry control networks and do not capture this change. We show an example of the trajectories in Figure 3, alongside with the mean deviation of the trajectory, and the conservation of angular momentum, energy and the prediction of kinetic and potential energy. Instead of predicting the deviation from circular orbits, we find that the HNN networks predict more or less circular orbits, which explains a relatively good conservation of angular momentum while predicting trajectories which are worse with respect to the true trajectory.

### 3.2.3 COUPLED HARMONIC OSCILLATOR

Again we keep the HNN architecture and the architectures in comparison to our previous experiments unchanged apart from adapting the input and output dimensions. As these experiments do not pose a large challenge for either HNNs and our symmetry control neural

---

[4]We check that there is no significant change in the fit accuracy when including higher order polynomials.

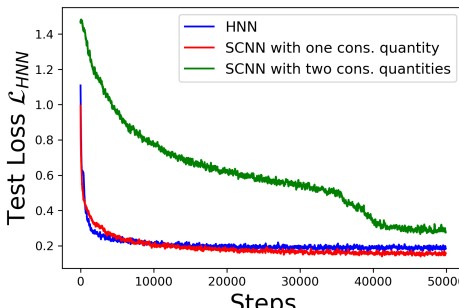

Figure 4: Comparing the test loss for HNN (blue), SCNN with one (red) and two (green) cyclic coordinates over the training time. We can see that too many conserved quantities lead to a dramatic drop in performance, while choosing the correct number of conserved quantities results in a similar performance than the HNN. We averaged over 100 time steps to get a smooth curve.

networks, we would like to focus on fitting analytic expressions for the conserved quantities and the Hamiltonian. For instance, in the case of $n = 2$, we find that the conserved quantities show explicitly the decoupling of the two sub-systems in our symmetry control networks. We find for the conserved quantities the underlying energies of the subsystem, e.g.:

$$P_1 = 0.67 \ q_1^2 - 3.6 \ q_1 q_2 + 8.23 \ q_2^2 + 0.46 \ p_1^2 - 1.78 \ p_1 p_2 + 4.2 \ p_2^2 \ (0.29),$$
$$P_2 = -4.53 \ q_1^2 - 1.74 \ q_1 q_2 - 0.9 \ q_2^2 - 4.5 \ p_1^2 - 1.88 \ p_1 p_2 - 0.57 \ p_2^2 \ (0.26), \qquad (17)$$

where the largest of the neglected terms arises with a factor smaller than 0.05 and the MSE of the fit is shown in the brackets. The entire Hamiltonian is also fitted very accurately as

$$H = 1.02 \ q_1^2 - 0.41 \ q_1 q_2 + 1.89 \ q_2^2 + 0.97 \ p_1^2 + 0.99 \ p_2^2 \ (0.01), \qquad (18)$$

where the largest term which is neglected has a factor smaller than 0.05 and the MSE error is shown in the brackets. For these experiments we have used a **SCNN-base** with two cyclic coordinates. Details regarding the data we have used for training and testing can be found in Appendix B.

### 3.2.4 Double Pendulum

The double pendulum is a good example to see what happens when looking at more conserved quantities than there exist. We tested the SCNN with one and two conserved quantities while setting the values of $\alpha_i = 1$ to increase the effect of our symmetry ensuring loss components ($\beta = 0$ for comparability). Comparing with the HNN, we find that the the test loss ($\mathcal{L}_{HNN}$) of the SCNN is very similar to the loss of the HNN for one conserved quantity, while for two conserved quantities the convergence is much slower and after 50000 steps the difference is still a factor three over the complete test set (see Figure 4). We are not able to find an analytical formular of the conserved quantity but this comes as no surprise due to the complicated structure of the Hamiltonian (cf. Appendix A).

### 3.2.5 Further Experiments and Summary of Performance

We have analysed the effect of varying the loss weights on our physical systems and find no large effects when varying the weights in the range $10^{-3} < \alpha < 1$ (cf. Appendix D for detailed results). A summary of the performance of our SCNNs in comparison to HNNs can be found in Table 1. To estimate the performance on the trajectories and the overall energy conservation we have integrated the mean square deviation from the true trajectory and the energy of the system respectively. Whereas the performance on the test loss is comparable to the HNN and baseline (a notable exception being the real harmonic oscillator and the double pendulum), the trajectories and energy are predicted much more accurately. An exception is the harmonic oscillator which already is very well captured with a HNN.

## 4 Related Work

There are several related directions where symmetries have been learned and utilised with neural networks. Also the use of physics bias not just via the Hamiltonian framework is playing a large role in related works. We briefly comment on several of them:

| Task | Test loss SCNN (HNN) | Trajectory SCNN (HNN) | Energy SCNN (HNN) |
|---|---|---|---|
| Two-body ($\times 10^6$) | $5.25 \pm 0.15$ ($2.13 \pm 0.04$) | $82e2 \pm 11e2$ ($114e3 \pm 9e3$) | $14.79 \pm 2.32$ ($19.12 \pm 1.92$) |
| Three-body ($\times 10^5$) | $6.85 \pm 1.96$ ($32.52 \pm 2.08$) | $14.0e2 \pm 3.4e2$ ($60.0e2 \pm 8.8e2$) | $77.1e2 \pm 55.6e2$ ($13.5 \pm 2.4$) |
| HO ($\times 10^6$) | $42.23 \pm 1.19$ ($8.49 \pm 0.23$) | $1.15e3 \pm 0.96e2$ ($2.29e2 \pm 1.39e2$) | $32.39 \pm 9.24$ ($8.75 \pm 2.88$) |
| Double Pend. | $24.46 \pm 2.00$ ($47.01 \pm 3.97$) | $13.2e2 \pm 5.0e2$ ($18.0e2 \pm 6.3e2$) | $18.6 \pm 5.6$ ($229.3 \pm 169.4$) |
| Real Sph. Pend. | $6.84 \pm 0.78$ ($6.61 \pm 0.75$) | — — | — — |
| Real HO($\times 10^0$) | $1.85 \pm 0.33$ ($6.10 \pm 1.83$) | — — | — — |

Table 1: Summary of quantitative results across tasks. All quantities are multiplied with $10^3$ if not otherwise stated.

- The baseline approach to predict dynamics has been studied in many other approaches previously (e.g. Grzeszczuk et al. (1998) for early work on this topic).

- An alternative to describing dynamics via Hamiltonian is via Lagrangian. The equivalent approach to HNNs in terms of learning Lagrangians is discussed in (Cranmer et al., 2020; Gupta et al., 2019; Lutter et al., 2019).

- Canonical transformations to cyclic coordinates have been utilised in the context of flow-networks. Bondesan & Lamacraft (2019) aim at finding the canonical transformation to cyclic coordinates but they do not study the improved predictions using those coordinates and their loss uses finite difference as opposed to an auto-differentiation loss we utilise here. Li et al. (2020) requires all coordinates to be cyclic (i.e. the system is integrable) which is not the case in our approach.

- Predicting the coordinates $(\mathbf{q}, \mathbf{p})$ from images has been pursued in (Greydanus et al., 2019; Toth et al., 2019). Here this would add an additional network before our input with the target output $(\mathbf{q}, \mathbf{p})$ as was explicitly demonstrated to work in (Toth et al., 2019).

- Very accurate prediction of dynamics can be performed using interaction networks (Battaglia et al., 2016; Watters et al., 2017). Biasing the local interactions via the Hamiltonian in graph networks has been discussed in Sanchez-Gonzalez et al. (2019). It would be very interesting to apply our approach in this framework to analyse the performance at larger systems.

- Obtaining analytic formulae for the network output on physical systems has been pursued in (Cranmer et al., 2019; Sahoo et al., 2018; Wetzel et al., 2020).

## 5 Discussion

One great challenge in physics is the search for conserved quantities and underlying symmetries. Our architecture provides an approach on how to find these symmetries using neural networks automatically without prior knowledge of them. We find interesting performance on simple toy systems where the symmetries are well known. We are able to identify these symmetries and by using them we can improve on performance with Hamiltonian neural networks.

It will be extremely interesting to extend on this proof of concept analysis by applying it to state of the art predictive networks such as in (Sanchez-Gonzalez et al., 2019) which as of now only utilise the Hamiltonian and not all of the symmetries.

A second line of investigation is in the realm of physics. An example for an area of future application is in integrable systems which for classical systems is satisfied if all coordinates are cyclic. Determining whether a system is integrable is generally speaking unknown and our method provides an automated route to search for the required structures.

We hope to report on progress in these directions at future conferences.

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

## A    ADDITIONAL SYSTEMS

This appendix presents details about some of the physical systems we have tested our SCNNs on.

### A.1    SPHERICAL PENDULUM

The spherical pendulum is governed by the following Hamiltonian:

$$\mathcal{H} = \frac{p_x^2 + p_y^2}{2m} - m \ g \ \sqrt{l^2 - q_x^2 - q_y^2} \ , \tag{19}$$

where we take a univeral mass $m = \frac{1}{2}$ coupling $g = \frac{1}{2}$ and pendulum length $l = 1$. Our data generation is described in Appendix B. In this system, the angular momentum in z-direction is conserved:

$$L = q_x p_y - q_y p_x \ . \tag{20}$$

Additionally, one can find the different Hamiltonians for the subsystems due to the fact that the spherical pendulum can be seen as a two-dimensional harmonic oscillator with degenerate eigenvalues.
We use the same architecture as for the harmonic oscillator, and focus on fitting conserved quantities. We enforce two cyclic coordinates (the maximal possible number) and scan over $\alpha_1 = \alpha_2 \in \{10^0, 10^{-1}, 10^{-2}, 10^{-3}\}$, and are able to extract the conserved quantities (using $\alpha = 10^{-3}$):

$$P_1 = 1.19 \ q_x^2 + 0.4 \ q_x q_y + 0.72 \ q_y^2 + 1.77 \ p_x^2 + 0.51 \ p_x p_y + 1.09 \ p_y^2 - 0.83 \ q_x p_y + 0.84 \ q_y p_x \ ,$$
$$P_2 = -0.65 \ q_x^2 + 0.58 \ q_x q_y - 1.38 \ q_y^2 - 0.96 \ p_x^2 + 0.77 \ p_x p_y - 1.92 \ p_y^2 - 1.18 \ q_x p_y + 1.18 \ q_y p_x \ , \tag{21}$$

with MSE of 0.003 and 0.002 and the omitted terms are below 0.05. One can see here the degenerated energies. The fitted Hamiltonian is:

$$H = 0.68 \ q_1^2 + 0.71 \ q_2^2 + 1.02 \ p_1^2 + 1.02 \ p_2^2 \ (0.03). \tag{22}$$

The true Hamiltonian (19) and the found one differ only by a factor of 0.005, and, therefore, linear regression finds a harmonic oscillator like structure.

### A.2    CHARGED PARTICLE IN A MAGNETIC FIELD

The charged particle in a magnetic field, where we choose a circular B-field plus an electric field, is governed by the following Hamiltonian:

$$\mathcal{H} = \frac{1}{2m} \left( (p_x + q \ q_y \ B)^2 + (p_y - q \ q_x \ B)^2 \right) + k \ \left( q_x^2 + q_y^2 \right) \ , \tag{23}$$

where we take a univeral mass $m = \frac{1}{2}$, electric field $k = 1$, charge $q = 1$ and B-field $B = 1$. Here, the angular momentum in z-direction is conserved

$$L = q_x p_y - q_y p_x \ . \tag{24}$$

We use the same neural network architecture as in the previous example, and enforce cyclic coordinate conditions on the maximum number of coordinates, but in this case we used $\alpha_1 = \alpha_2 = 10^{-2}$ due to the better performance when fitting the analytic quantities. Using linear regression, we find the following analytic expressions for the conserved quantities and for Hamiltonian:

$$P_1 = 3.59 \; q_x^2 + 3.58 \; q_y^2 + 1.80 \; p_x^2 + 1.79 \; p_y^2 - 2.74 \; q_x p_y + 2.75 \; q_y p_x \; (0.43),$$
$$P_2 = -2.33 \; q_x^2 - 2.32 \; q_y^2 - 1.17 \; p_x^2 - 1.17 \; p_y^2 + 2.97 \; q_x p_y - 2.96 \; q_y p_x \; (0.32),$$
$$\mathcal{H} = 2.00 \; q_x^2 + 2.00 \; q_y^2 + 1.00 \; p_x^2 + 1.00 \; p_y^2 - 2.01 \; q_x p_y + 2.01 \; q_y p_x \; (0.01), \qquad (25)$$

where we omitted terms smaller than 0.05 and the numbers in the bracket indicate the MSE of the fit.

### A.3 Real harmonic oscillator

As in the previous cases, we use the SCNN architecture to predict the motion of the particle. We are limited to the given data points from the datasets, and therefore only have one given trajectory. Therefore, the SCNN is not able to find the subsystems of the two dimensional harmonic oscillator, but is able to act on the test set with a good accuracy and performs much better than the HNN (see Table 1).

## B  Data

### B.1 Two-body problem

As described in the main text we set both masses and the coupling to unity. We sample 1000 trajectories with 50 data points each over a time span of 20 which corresponds of at least two third of a full circle. We uniformly sample the starting point for the first body $\mathbf{q}_1$ from the box $[0.5, 1.5]^2$. The starting position for the second body is at $\mathbf{q}_2 = -\mathbf{q}_1$.

We choose the velocity of both bodies in such a way that the bodies describe near circular orbits and we give the entire system an overall momentum $\mathbf{p}_{\mathrm{com}} = (\epsilon_1, \epsilon_2)$

$$\mathbf{p}_1 = (1 + \epsilon) \; \mathbf{p}_{\mathrm{circ}} + \mathbf{p}_{\mathrm{com}} \; ,$$
$$\mathbf{p}_2 = -\mathbf{p}_1 + 2 \; \mathbf{p}_{\mathrm{com}} \; , \qquad (26)$$

where $\epsilon$ and $\epsilon_{1,2}$ are sampled from a normal distribution with $\sigma = 0.05$ and 0.1 respectively.

We take 80 percent of the data for training and use the rest for testing.

### B.2 Three-body problem

The setup is relatively similar to the two-body problem as we consider three body near circular motions and give the overall system an overall momentum. We initialise the three particles on the corners of an equilateral triangle. The velocity of the first particle is given as follows

$$\mathbf{p}_1 = (1 + \epsilon) \; \mathbf{p}_{\mathrm{circ}} + \mathbf{p}_{\mathrm{com}} \; , \qquad (27)$$

where the remaining velocities are chosen accordingly as in the two-body problem and where the center of mass motion and the deviation from the circular motion are exactly generated as in the two-body example. We use 5000 trajectories with 20 data points for training and testing of our networks. The time span is 5 which corresponds to motion which is mostly circular. When using longer time spans, we find that convergence of the networks becomes problematic.

### B.3  $n$-dimensional harmonic oscillator

The underlying Hamiltonian (14) is generated as follows. The coupling matrix is generated as the product of an orthogonal matrix $S$ and a diagonal matrix $D$ as $A_{ij} = SD^2 S^T$.

The entries of the diagonal matrix $D$ are random entries sampled uniformly in the interval $[1 - \sqrt{10}]$ which are then rounded to one digit to ensure that the values are *sufficiently rational*, i.e. that the motion is periodic. For instance, if we choose an entry in $D^2$ equal to 2 the motion is not periodic whereas if we choose 1.96 the motion is periodic. The matrix $S$ is obtained by generating random normal distributed entries (mean 0 and standard deviation 0) and we then orthonormalize the vectors.

The initial conditions $(\mathbf{q}(t = 0), \mathbf{p}(t = 0))$ are a $2n$-dimensional vector which we generate as follows. We generate a vector with random entries (uniform sample) in the interval $(0 - 2)$. We then normalize the vector to unit length and then re-scale it uniformly to a vector of length in the interval $(0.2, 2.0)$.

For our dataset we utilise 100 initial conditions and generate 500 points on the trajectory. These points are taken from a time span of 10 which corresponds to a time where multiple periodic motions are included. We split it into equal size training and test set.

### B.4 SPHERICAL PENDULUM

As described above we fix all parameters, and sample the starting conditions $(\mathbf{q}(\mathbf{t = 0}), \mathbf{p}(\mathbf{t = 0}))$ from a uniform distribution from the interval $(-1, 1)$. We ensure, that the energy is well-defined and negative.

For our dataset we utilise 100 initial conditions and generate 500 points on the trajectory. These points are take from a time span of 10 which corresponds to a time where multiple periodic motions are included. We use a 80:20 split for the training and test set.

### B.5 CHARGED PARTICLE IN A MAGNETIC FIELD

As described above we fix all parameters, and sample the starting conditions $(\mathbf{q}(\mathbf{t = 0}), \mathbf{p}(\mathbf{t = 0}))$ from a uniform distribution from the interval $(-1, 1)$, We then normalize the vector to unit length and then re-scale it uniformly to a vector of length in the interval $(0.1, 1.0)$. For our dataset we utilise 100 initial conditions and generate 500 points on the trajectory. These points are take from a time span of 10 which corresponds to a time where multiple periodic motions are included. We use a 80:20 split for the training and test set.

### B.6 DOUBLE PENDULUM

As described in the main text we fix all parameters, and sample the starting conditions $(\mathbf{q}(\mathbf{t = 0}), \mathbf{p}(\mathbf{t = 0}) = \mathbf{0})$ from a uniform distribution from the interval $(0, 2\pi)$. Note, that $\theta_1$ and $\theta_2$ are angles and therefore, we have to ensure that they always lie in the interval $(0, 2\pi)$. For our dataset we utilise 100 intial conditions and generate 500 points on the trajectory. These points are take from a time span of 10 which corresponds to a time where multiple periodic motions are included. We use a 80:20 split for the training and test set.

### B.7 REAL HARMONIC OSCILLATOR

In this real world example we use data from the paper Schmidt & Lipson (2009), namely the example which is called „two-masses-three-springs". To measure the data points, they use a system with two fixed boundary points, connected by a system with three springs and two masses. As this corresponds to two real harmonic oscillators we refer to this system as real harmonic oscillator. The dataset consists of one trajectory from a real world system which is loosing energy. To make it comparable to the paper of Greydanus et al. (2019), we used the same split (taking the first 80 percent as training set, and the remaining part as test set). Therefore we have a different distribution in the training set and the test set.
Because the data consists of only a single trajectory of data points we are technically not able to find a conserved quantity. Therefore, we just test the performance and check how well the network generalizes to the test set.

| | **Two-body MSE Trajectory** | | | |
|---|---|---|---|---|
| $\alpha_2$ / $\alpha_1$ | $10^0$ | $10^{-1}$ | $10^{-2}$ | $10^{-3}$ |
| $10^0$ | $7.41 \pm .66 \times 10^0$ | $9.87 \pm 1.16 \times 10^0$ | $6.06 \pm 0.61 \times 10^{-1}$ | $3.30 \pm 0.70 \times 10^{-1}$ |
| $10^{-1}$ | $2.88 \pm 0.23 \times 10^1$ | $1.58 \pm 0.08 \times 10^1$ | $1.34 \pm 0.11 \times 10^{-1}$ | $1.83 \pm 0.32 \times 10^{-1}$ |
| $10^{-2}$ | $3.00 \pm 0.48 \times 10^0$ | $2.25 \pm 0.48 \times 10^{-1}$ | $8.59 \pm 2.39 \times 10^{-1}$ | $4.14 \pm 0.28 \times 10^{-1}$ |
| $10^{-3}$ | $4.49 \pm 0.30 \times 10^1$ | $5.94 \pm 2.51 \times 10^{-1}$ | $1.67 \pm 0.25 \times 10^{-1}$ | $1.01 \pm 0.12 \times 10^{-1}$ |

Table 2: Loss-hyperparameter scan for the two-body problem. We show the integrated MSE of the predicted and true trajectories.

### B.8 Real Spherical Pendulum

In this real world example we use data from the paper Schmidt & Lipson (2009), namely the example of the real pendulum. Due to the fact that we want to have more than one degree of freedom (otherwise the only conserved quantity is the energy), we use always a pair of two data points from the one-dimensional pendulum to get a two dimensional movement which is physically equivalent to a spherical pendulum. We use the first 80 percent of the data points as training set, and the remaining ones as test set. Because this is a real-world system it looses some energy, and therefore, we have a slightly different distribution on the training set and the test set, but this difference is no limitation for the search of conserved quantities.

## C Hyperparameters

We have performed some hyper-parameter searches for our symmetry control networks on which we give an overview here.

By varying the hidden layer size between 50-300, we find that there is a minimum size of layer size 100 to find convergence of our networks. We have varied the number of hidden layers up to 5 hidden layers. Two hidden layers are already sufficient (tested up to 5 hidden layers), hence we restricted ourselves on them. Coarsely speaking, we find that the precise architecture is less relevant in our current experiments.

More relevant are the pre-factors in the loss. Depending on the choice, we can either force our networks for better performance on the predictions or obtaining the conserved quantities. For this trade-off, we have optimized the experiments in this paper on the particle trajectories.

We find that pytorch's standard orthogonal initialization provides the best results out of the standard initializations. We have not observed a large random seed dependence.

## D Loss Prefactors

In our analysis of the loss factors we consider SCNNs with maximum number of conserved quantities. This leaves us with two loss hyperparameters $\alpha_1$ and $\alpha_2$ which we analyze here. We scanned them over the values $\{10^0, 10^{-1}, 10^{-2}, 10^{-3}\}$ for the 2-body problem. The results are in Table 2 below, where you can see the average error of the trajectories as used in the main text. As a comparison, the error of the HNN is significantly higher with $1.15 \pm 0.09 \times 10^{-1}$.

The second measure we compare are the conserved quantities. Therefore, we look at the conservation of energy and angular momentum which is shown in Table 3 and 4. As a comparison, the error of the energy for HNN is $1.91 \pm 0.19 \times 10^{-5}$ and $1.23 \pm 0.10 \times 10^{-3}$ for the angular momentum.

| Two-body deviation energy | | | |
|---|---|---|---|
| $\alpha_2$ 
 $\alpha_1$   $10^0$ | $10^{-1}$ | $10^{-2}$ | $10^{-3}$ |
| $10^0$   $2.20 \pm 0.29 \times 10^{-1}$ | $2.04 \pm 1.38 \times 10^0$ | $4.83 \pm 4.74 \times 10^0$ | $2.01 \pm 0.77 \times 10^{-5}$ |
| $10^{-1}$   $2.88 \pm 0.23 \times 10^1$ | $1.68 \pm 0.56 \times 10^0$ | $8.80 \pm 1.77 \times 10^{-5}$ | $5.40 \pm 1.04 \times 10^{-6}$ |
| $10^{-2}$   $2.43 \pm 0.28 \times 10^{-1}$ | $8.75 \pm 3.47 \times 10^{-3}$ | $6.01 \pm 3.38 \times 10^{-4}$ | $3.50 \pm 0.48 \times 10^{-5}$ |
| $10^{-3}$   $1.87 \pm 0.38 \times 10^{-1}$ | $3.34 \pm 2.61 \times 10^{-3}$ | $5.49 \pm 0.54 \times 10^{-5}$ | $1.13 \pm 0.20 \times 10^{-5}$ |

Table 3: Loss-hyperparameter scan for the two-body problem. We show the integrated deviation of the energy between the predicted and true trajectories.

| Deviation of Angular Momentum for Two-body system | | | |
|---|---|---|---|
| $\alpha_2$ 
 $\alpha_1$   $10^0$ | $10^{-1}$ | $10^{-2}$ | $10^{-3}$ |
| $10^0$   $1.69 \pm 0.21 \times 10^1$ | $1.70 \pm 0.15 \times 10^0$ | $2.42 \pm 2.37 \times 10^{-1}$ | $7.22 \pm 3.82 \times 10^{-3}$ |
| $10^{-1}$   $1.33 \pm 0.18 \times 10^2$ | $1.08 \pm 0.35 \times 10^2$ | $4.74 \pm 1.28 \times 10^{-3}$ | $7.81 \pm 2.15 \times 10^{-4}$ |
| $10^{-2}$   $5.08 \pm 0.60 \times 10^1$ | $5.95 \pm 2.37 \times 10^0$ | $4.94 \pm 2.78 \times 10^{-1}$ | $1.64 \pm 0.31 \times 10^{-4}$ |
| $10^{-3}$   $1.16 \pm 0.15 \times 10^2$ | $4.15 \pm 3.00 \times 10^0$ | $3.74 \pm 0.74 \times 10^{-3}$ | $2.97 \pm 0.63 \times 10^{-4}$ |

Table 4: Loss-hyperparameter scan for the two-body problem. We show the integrated deviation of the angular momentum between the predicted and true trajectories.

## E   Power for Multinomial-Expansion

In this appendix we elaborate how we decide which power for the multinomial expansion. For illustration purposes, we use the two-body example here.

We use the test set as the coordinates and fit a multinomial function using LinearRegression and PolynomialFeatures implemented in sklearn (Pedregosa et al., 2011). When looking at the MSE-plot for the different degrees (cf. Figure 5), one can directly find a cut-off to determine the correct degree. Coordinate 1 and Coordinate 2 have degree one (both give us momentum conservation), while the fourth one can be fitted by a multinomial of order two (It is equivalent to the angular momentum). On the other hand, one can directly see, that Coordinate 3 has no multinomial expression, the error is small, but does not converge to a specific value.

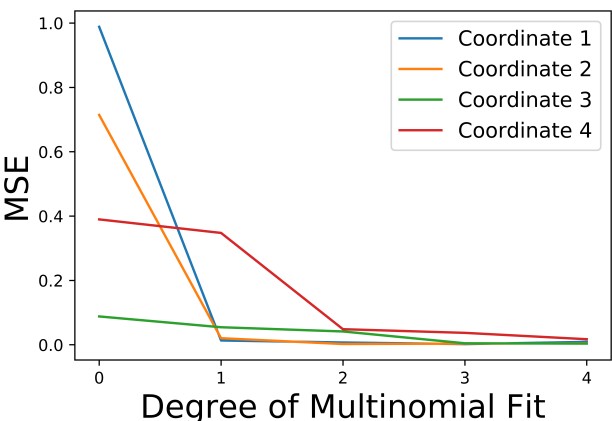

Figure 5: Comparing the MSE depending on the degree of the used polynomials. We can see a drastic drop of the error for a specific degree. When looking at the linear regression prefactors, we can identify the conserved quantity. The only distinct behaviour of the MSE is for coordinate 3. Here, we are not able to find a suitable fit due to the problem, that we are not able to fit the Hamiltonian function for the 2-body-problem with polynomials.

