# OpenReview forum: "Symmetry Control Neural Networks"
_ICLR.cc/2021/Conference — Reject_

### Official Review · AnonReviewer1 · 2020-10-27
**Interesting research on finding conservative quantities to improve prediction, however not yet convincing**

**Rating:** 5
**Confidence:** 3

**Review:**

The paper proposes a set of loss functions that can make neural networks learn Hamiltonian dynamics with conserved quantities.
The research builds on Hamiltonian networks and adds the search for conserved quantities as an auxiliary task.  The loss functions are a straight forward translation of equations describing Hamiltonian dynamics with conserved quantities.

For the machine learning audience, the paper could be written more didactically, such that it can be easier followed. In some parts, information is missing or misleading and the graphics need some improvements. Otherwise, the presentation is adequate.

The proposed method is novel, as far as I can tell. Although fascinating, its impact will probably not be very high, as real-world systems, for instance, robot dynamics are not a closed system and thus energy conservation is not given. Nevertheless, I would love to see more progress in this direction.

My main concern with the paper, as it is mostly empirical, does not provide enough evidence that the method would work in non-toy settings.

Concrete problems:
- the number of conserved coordinates has to be known
    - here I think it would be possible to show what happens when the number of conserved quantities is scanned through. I guess a breakdown in performance would be observed when the number is too high. I think something like this would strengthen the paper

- the different models are confusing: If I understand correctly, then Model 1 is your method in its general form and is the only real model to be compared to. It only has the knowledge about how many conserved quantities are the system. The other models are some kind of ablation, an additional way to check which parts perform how well. I appreciate those models, but they should not be called model 1...5 but rather Sym-Net and Sym-Net + Oracle X Y Z, because they get quite a bit of extra information.

- already for 3 bodies the new network does not perform much better (sometimes worse) than the baseline on its own terrain -- the conserved quantities:
    - after 8 steps HNN is better or equally good on all conserved quantities (Fig 3). Still, the prediction is a bit better than the HNN, which puzzles me a bit. What is the reason for the better performance? Also, the qualitative predictions in Fig 3 are not really convincing.
- using data from a non-synthetic dataset that you are not generating yourself would strengthen the paper

As you are also after finding concise analytical expressions, your approach could be potentially nicely combined with the "equation learning" architecture [1]. In any case, it might also be a good baseline. It would be valuable to know whether the inductive bias of a functional form or the conserved quantities is more effective.

Details:
 - below Eq 3: unpack and explain a bit why the {conserved quantity,H} vanishes. (I think this will increase readability)
 - Eq 4: maybe Q and P could be introduced before they appear in EQ 4.
 - page 3: 2. state what you need from the Poisson algebra here: {p_i,p_j} = 0,  {q_i,q_j} = 0, {p_i,q_j} = \delta_ij
- p4 check last sentence
- Fig 2 and 3 Font size is an order of magnitude to small. A proper legend would also help.

[1] http://proceedings.mlr.press/v80/sahoo18a.html

--- Post rebuttal update

The authors clarified and removed problems in the paper and did additional experiments. The changes to the paper are quite substantial and I cannot make a full review again.  (Upvote from 4->5)

---

> ### Author Response · Authors · 2020-11-23
> **Comment for Reviewer 1**
>
> We would like to thank the referee for the insightful comments on our paper. Below we briefly comment on them and highlight our changes in the new version of the paper (a concise summary of all major changes is included as a separate comment):
>
> With respect to readability we have stream-lined the presentation of our experiments, improved the readability of the figures, and added a figure showing the effect of our additional loss component.
>
> We have added two examples where we test our setting on real experimental data. The data is such that the system is loosing energy (due to friction) and we find that our SCNN does work in this setup. Interestingly, we observe that the test loss is even below the test loss compared to the HNN which we do not observe in synthetic examples. We think that this addresses your concern about applicability of our method on non-toy settings.
>
> Below are comments regarding the concrete problems:
>
> - Number of conserved quantities: In all cases (now referred to as SCNN-base) we actually exhaust the list of conserved quantities, i.e. we required the maximum number of conserved quantities given by the half the dimension of phase space. To illustrate the breakdown in performance for systems with fewer conserved quantitites, we have added the classic example of a double pendulum in the revised version where we see a breakdown in performance when searching for two conserved quantities in comparison to the improvement over HNN when looking at just one conserved quantity (Figure 4). We think that this example illustrates nicely, that one should in principle proceed iteratively if one does not see good performance with an exhaustive list.
> - Confusion about different models: we hope that our re-writing of section 3 now states clearly our intentions of the experiments. The main motivation behind our experiments is to see 1) that our method is working (now called SCNN-base), 2) when we add domain knowledge about conserved quantities we can improve performance using our loss constraints, and 3) our method produces interpretable formulae for the conserved quantities.
> - We have now added non-synthetic data and find that our method is still working.
> - We have added a reference to this interesting equation-learner paper in the related works section.
>
>
> Regarding the detailed points, here is a summary of our changes:
>
> - Below eq 3: Just to clarify, a conserved quantity on phase space is a function whose time derivative vanishes. Eq.3 tells us that the time derivative of any function we are interested in is given as the Poisson bracket of that quantity with the Hamiltonian. We have added a small explanation which hopefully increases readability. Also we think it is helpful to visualize the constraint and show in Figure 1 the explicit constraint of the conservation of angular momentum on the model parameters in a simplified 2-parameter example of the 2-body problem.
>
> - Eq 4: The definition of Q and P is given in the first line of Equation 4, i.e. how these new coordinates P and Q are obtained from the old ones via the map T which satisfies the properties as in the second line of Equation 4.
> To make it more clear, we added $T$ in the sentence above Equation 4.
>
> - page 3: Poisson algebra. We have added a sentence explaining where this loss component is used: We need the second part of the Poisson loss to ensure that our transformation T_\psi is of the correct type. The first part {P,Q}=\delta ensures that we do not learn the same conserved quantity twice and that the trivial (vanishing solution) is not allowed.
>
> - page 4: thank you for spotting the typo which is no longer present due to the restructuring of Section 3.
>
> - We have now adapted the font sizes in the figures.
>
>
> We think that these changes address your points and have improved the quality of our paper.

---

### Official Review · AnonReviewer2 · 2020-10-28
**an exploration of learning symmetries in the HNN paradigm**

**Rating:** 5
**Confidence:** 4

**Review:**

This submission explores the question of identifying conserved quantities in Hamiltonian dynamics for physical systems by attempting to learn canonical transformations. The approach closely resembles previous work on "Hamiltonian neural networks" but the loss is augmented with a term enforcing the invariance of the dynamics under the transformation and with a term that ensures the resulting transformed coordinates satisfy the constraints of the algebraic relations that emerge from the Poisson bracket. Together these two additions allow the authors to train a network that performs a change of coordinates which is subsequently optimized to bring it closer to a canonical transformation. Perhaps the main observation is that some of the cyclic coordinates identified by the network have a clear relation to the underlying conserved quantities.

On the whole I was not particularly impressed by the numerical results---despite imposing fixed numbers of conserved quantities, the performance is not meaningfully superior to the established HNN framework. I did not understand why the authors would use an integrator that is not symplectic---this seems like a clear route to bettering the performance over longer timescales. What is more, the weight of the additional terms in the loss was not systematically varied, so was not left with much sense of the extent to which it was affecting the optimization.

---

> ### Author Response · Authors · 2020-11-23
> **Comment for Reviewer 2**
>
> We would like to thank the referee for the comments and suggestions on our paper. Beyond our general reply on the changes of our paper, we would like to comment on your points below:
>
> With respect to the numerical results, we would consider the order of magnitude improvement compared to HNN in terms of the deviation of the trajectories (e.g. in the 3-body problem) and the qualitative capturing of key aspects in the motion as significant. As can be seen from our new summary table, we reach an improved performance throughout many examples, including in particular also examples which are trained on real experimental data.
>
> The other aspect which is of significant interest is to find the conserved quantities in a system. This typically requires domain knowledge. Here we show how to circumvent this problem and to find interpretable quantities without knowing them in principle.
>
> - Symplectic integrator:
> Clearly, for actual integration a symplectic integrator is superior to a Runge-Kutta method. However, the problem we are solving here is a different one: we are looking for finding the Hamiltonian in the first place which we want to use for the integration in a second step. The comparison of the performance is done using a standard Runge-Kutta integrator to highlight the differences between a baseline architecture and our modified HNN architecture. A baseline architecture does not have a Hamiltonian description and cannot be used with a symplectic integrator. To give a rough estimate on the effect of using a symplectic integrator and our standard integrator, we have compared the predictions we get from using the accurate masses and slightly modified masses to mimic an inaccurate Hamiltonian. This deviation is significantly larger than the change in the accuracy of the integrator for which we used the implementation in the publicly available Rebound package. We have added a comment at the end of Section 2 to clarify this point about the integrators.
>
> - Loss weights:
> We have added more information on the impact of our additional losses. In particular, we have highlighted in Section 2 how additional loss terms constrain the network parameters by visualising the constraint arising from the conservation of angular momentum in the 2-body problem (Figure 1). We think that this provides a nice visualization of the role our Poisson loss has on the model space. We have also added a description of what is happening when we vary the individual loss parameters which can be found in Appendix D. We find that there is large range of values for \alpha_i where we find good results. In the main text we mention this result in the summary of our results.
>
> We think that these clarifications to your points have improved our paper.

---

### Official Review · AnonReviewer3 · 2020-10-29
**Review of Symmetry Control Neural Networks for ICLR 2021**

**Rating:** 5
**Confidence:** 5

**Review:**

**Summary.** The authors propose using a neural network to learn a canonical transformation of the data coordinates before learning a Hamiltonian. This is a novel contribution in that previous work has shown how to learn Hamiltonians with neural networks but it has not shown how to learn the proper canonical transformation. In the course of learning this canonical transformation, they show how to project out other symmetries (linear and angular momentum) and hence improve upon HNNs while also learning these other symmetries to a good approximation.

**Strong points.** The authors have a strong grasp of Hamiltonian mechanics and they lay out the theory in an accurate and easy-to-follow manner. The core idea is a good one and it leads to promising (although entirely qualitative) results. The experimental setup is reasonable and the behavior of the trained models is visualized effectively. The symbolic regression applied to the learned models shows that they learned physically relevant quantities, and this is an excellent empirical validation of the authors’ approach, eg. Equation 16.

**Weak points.** Some sections of the paper were very hard to follow. For example, the experimental setup which was introduced in the form of “Model 1, Model 2, Model 3…” was very confusing. First of all, the authors never gave intuition for what each of these models was specifically aiming to test. Here are my best guesses, although I would like to see the authors’ definitions as well:
* Model 1: This appears to be the generic Symmetry Control NN. They enforce a Poisson loss in latent space and then train an HNN on it. Since they set \beta to zero, H ends up being invariant to the transformed coordinates (P, Q) only implicitly, by way of the Poisson loss and the HNN loss. Since the fourth loss term is regularizing P_i to be constant, it appears that they are regularizing *all* P_i in the model to be constant. I don’t quite understand this. This presumes that all momenta are stationary, which is not the case for the 2-body system -- we only expect two of them to be constant. Is the idea that this regularization will hopefully force two to be constant while the other two are not?
* Model 2: This enforces only half of the transformed coordinates to be cyclic. But the authors make another change: they force the learned canonical transformation to be linear by construction. So to summarize, they are changing two things: 1) now we're using a linear transformation for (p,q)->(P,Q) and 2) now we're using a different number of cyclical coordinates. A proper ablation of the experiment should just change one of these, and the authors should describe what they are trying to ablate.
* Model 3: This resembles Model 1 more closely, but here we are explicitly defining the first two symmetries (x and y momentum conservation) but not the third (angular momentum)
* Model 4: This is complementary to Model 3 in that they are fixing the angular momentum and learning the linear momenta
* Model 5: They fix all the known symmetries (both angular and linear momenta). This corresponds to adding maximal domain knowledge to the model.
The remainder of the experimental results likewise need more explanations. There should be a table of quantitative results. There should be more discussion regarding the system of coupled oscillators. There should be quantitative measurements of error-of-fit for the symbolic regression models
	The 3-body results are quite good and it was interesting to see that these models excelled in this context compared to baseline NNs and HNNs. The authors provide a nice, intuitive discussion of these results.

**Recommendation.** 5 : Marginally below acceptance threshold

**Reasoning.** This paper tackles a significant problem, presents a novel and useful method, and achieves promising empirical results. Its weaknesses were 1) that experimental methods and results were not sufficiently well explained and 2) there were not enough quantitative results. The paper cannot be accepted to ICLR as-is. I would consider changing my recommendation if these two core issues were addressed in a substantial way.

**To improve the paper.** The authors should make their explanation of methods substantially clearer, with special attention paid to explaining why they design the experiments and models the way they did. The authors should provide qualitative results for all three tasks.

---

> ### Comment · AnonReviewer3 · 2020-11-23
> **General update**
>
> I do not see any responses from the authors. I have read the other reviews and generally agree with them. I continue to believe that this paper tackles a useful and significant problem. However, given its weaknesses (described in original review) I will maintain my score.

---

> ### Author Response · Authors · 2020-11-23
> **Comment for Reviewer 3**
>
> We would like to thank the referee for the comments and suggestions on our paper.
>
> We have changed the discussion of our experiments, to clearly state our intentions behind our different models. There are three objectives we would like to address with our experiments: 1) [SCNN-base] experiments shall establish whether our SCNN architecture performs well in comparison to HNN and baseline architectures. 2) [SCNN-constraint] are models where we include additional domain knowledge about conserved quantities. We want to find out whether this can improve on our prediction for the Hamiltonian and subsequently the dynamics. For instance, in the three-body example this improvement is quite large for the SCNN-constraint models. 3) Fits of [Analytic formulae] for the cyclic coordinates which allow us to find excellent agreement with known conserved quantities in these systems.
>
> During this streamlining of our discussion we have removed Models 2 and 3 in the 2-body case from the current form of the paper to avoid confusion. They were intended to provide even better fits on the conserved quantities. However, our previous Model 1 (now called SCNN-base model) already produces very accurate expressions for the conserved quantities and for clarity we use the prediction of analytic formulae we obtain from this model.
>
> We have added errors to the fit of the symbolic regression models in the revised version and we have provided Appendix E which includes more details about our fitting of symbolic formulae.
>
> We have added several further examples including real experimental data where we find that our SCNNs are working. In particular, these additional examples include several setups which are different versions of harmonic oscillators (spherical pendulum in the appropriate limit, two-masses-three spring real data system). Our double pendulum example clarifies that our method can be used to estimate the number of conserved quantities as we observe significant worse performance when demanding too many conserved quantities.
> We provide a summary table to give an overview of our findings in comparison to HNN in Section 3.
>
> We think that with these changes we explain our experimental results much clearer and that our additional experiments provide significantly more quantitative results. Beyond these changes that address your comments, we have made several other changes to improve the readability and to address further points with respect to our experiments. A summary can be found in our general reply.

---

### Official Review · AnonReviewer4 · 2020-10-29
**an application of NN to learning symmetries in physics**

**Rating:** 4
**Confidence:** 4

**Review:**

This paper presents the results of a NN trained to learn symmetries in physics, specifically, to learn and preserve quantities that are preserved (e.g., energy, angular momentum). The input is a sequence generated from a Hamiltonian dynamics. Results of experiments on 2 and 3 body problems and a harmonic oscillator are presented. The training networks are small, shallow feedforward networks. There is some customization of the training networks to incorporate "cyclic" coordinates. Results indicated empirical conservation up to small error of physically conserved quantities. The paper is fairly easy to read, with much relevant background provided.

In the early days of NNs, this might have been a very interesting paper. With today's advances, and NN finding success in almost every area with data, it is not clear what the contribution of this paper is. Perhaps the main innovation is the design of the networks. Unfortunately, there is little explanation provided of the experimental results.

-- Why is this result interesting? Given that the output is a simple function of the input, why is this result surprising in the least?

-- Given that the model for data is explicit and the training model is simple, can you say anthing rigorous to explain the results obtained empirically?

-- the outputs are close but do not perfectly periodic. What parameter/model changes could possibly explain this? Would a different representation do better?

---

> ### Author Response · Authors · 2020-11-23
> **Comment for Reviewer 4**
>
> We would like to thank the referee for the comments and suggestions on our paper. Let us comment below on the questions of the referee:
>
> - Why is this result interesting? The results are interesting for two reasons:
> 1) Our result shows that we can find analytic formulae for conserved quantities underlying the dynamical system which are in very close resemblance of the actual conserved quantities known in physics. This is of large interest in fundamental physics where we often do not know the underlying conserved quantities/symmetries of a system. This is also of interest for any task where the symmetries of the system shall be learned from scratch.
> 2) The introduction of physics bias, in particular as shown here by using the constraints arising from symmetries, improves performance in comparison to baseline architectures (cf. our new summary table in our experiments section). This approach has been adopted by numerous works following the HNN work which appeared last year at NEURIPS. Our approach allows to find such symmetries as the HNN approach was able to find the Hamiltonian. Our approach places additional constraints on the allowed functions for the Hamiltonian. We have highlighted these points in the revised version by adding additional explanation for our loss function, in particular the constraints placed by the conservation of angular momentum in the two-body problem (cf. Figure 1).
>
> - Rigorous explanations of the results?
> We are adding additional loss components which are minimized in the presence of appropriate symmetries. This puts an additional constraint on the function space which effectively biases our network to a better solution. Although we do not provide a formal proof (we wanted to experimentally see this effect first), we have added a toy example in the case of the 2-body example where we can easily visualise the constraint from conserved quantities (Figure 1). We hope to increase the readibility of our article and to provide the reader with an intuition why this physics formalism provides stronger constraint on function space.
>
> - Perfectly periodic outputs.
> The models we have chosen are in close resemblance to the HNN framework where in fact most systems are showing periodic behaviour as they are integrable. In non-integrable systems such as the double pendulum, which we now include in our experiments, the motion is no longer periodic. In the former examples, the outputs are periodic in two different ways: 1) the almost circular motion visible in (q,p). 2) when going to appropriate coordinates, in particular when coordinates become our cyclic coordinates, the description becomes even simpler. We are in fact seeking such a representation with our cyclic coordinates. When physical systems are described in these cyclic coordinates, the Hamiltonian is independent of the Q-coordinates and the P-coordinates are constant in time. Our impression is that it is very worth studying and utilising these representations as they allow a very compressed description of the dynamical system. In our mind this is yet another reason why our results are interesting because they allow to connect to this compressed description of dynamical systems.
>
> We think that these modifications improve our paper. Further changes to improve the presentation and to strengthen our numerical results are summarised in our general reply.

---

### Author Response · Authors · 2020-11-23
**General Comment to Version2**

Based on the replies of the reviewers, we have improved the presentation of our paper as outlined below. We also provide detailed comments on the individual points of the reviewers as replies to their respective comment. This comment is to summarise the main changes in version2:


- We have added clarifications in our theory part to provide the reader more intuition of our additional losses (e.g. clearly showing that our additional loss provides meaningful constraints on the model space, cf. Figure 1).

- We have streamlined the description of our experiments. We start with a clear statement on what our intentions are (1. checking the performance of our novel loss/architecture approach. 2. Whether we can achieve improved performance by providing domain knowledge of known conserved quantities. 3. To find analytic formulae for conserved quantities) and subsequently discuss the individual experiments before providing a summary performance table.

- We have added some physical systems and systems with publicly available experimental data to highlight that our method successfully works on real data and is not limited to synthetic data, and that we can algorithmically identify the number of conserved quantities without knowing them in the case of the double pendulum (using synthetic data).

- We have provided a detailed hyperparameter scan on the loss weights to experimentally estimate their respective influence.

This clarifies our experimental findings that our method can improve on HNN and baseline architectures, and most importantly - in our mind - opens the door to search and identify conserved quantities of a system.

Beyond these major changes we have addressed smaller points in the individual replies for each referee.

---

### Decision · Program_Chairs · 2021-01-07
**Final Decision**

**Decision:**

Reject

**Comment:**

This paper proposes to learn symmetries of a physical system jointly with its Hamiltonian from data by learning a canonical transformation that render some of the coordinates constant.
The Hamiltonian dynamics and "canonical" transformation are softly enforced via loss terms.
A few experiments are performed demonstrating that the idea works and can learn a few approximate invariants, as well as some improvements over baselines agnostic to the symmetries.
The idea is interesting, but the experiments are limited in scope. It is not clear how to extend this idea to more complex systems where we do not know the number of conserved quantities in advance. It is also not clear how good are the learned invariants, as the results showing errors in conserved quantities (Fig 3) suggest that it is not very precise beyond a few time steps.